# Identification of Key Differentially Expressed mRNAs, miRNAs, lncRNAs, and circRNAs for Xp11 Translocation Renal Cell Carcinoma (RCC) Based on Whole-Transcriptome Sequencing

**DOI:** 10.3390/genes14030723

**Published:** 2023-03-15

**Authors:** Changqi Deng, Chengcheng Wei, Yaxin Hou, Ming Xiong, Dong Ni, Yu Huang, Miao Wang, Xiong Yang, Ke Chen, Zhaohui Chen

**Affiliations:** 1Department of Urology, Union Hospital, Tongji Medical College, Huazhong University of Science and Technology, Wuhan 430000, China; 2Department of Urology, Tongji Hospital, Tongji Medical College, Huazhong University of Science and Technology, Wuhan 430000, China

**Keywords:** bioinformatics, ceRNA, RCC, whole-transcriptome sequencing

## Abstract

We carried out whole transcriptome sequencing (WTS) on the tumor and the matching adjacent normal tissues from five patients having Xp11 translocation renal cell carcinoma (RCC). This was performed in terms of obtaining more understanding of the genomic panorama and molecular basis of this cancer. To examine gene-regulatory networks in XP11 translocation RCC, variance expression analysis was carried out, followed by functional enrichment analysis. Gene Expression Omnibus (GEO) of Xp11 translocation RCC data was used to validate the results. As per inclusion criteria, a total of 1886 differentially expressed mRNAs (DEmRNAs), 56 differentially expressed miRNAs (DEmiRNAs), 223 differentially expressed lncRNAs (DElncRNAs), and 1764 differentially expressed circRNAs (DEcircRNAs) were found. KEGG enrichment study of DEmiRNA, DElncRNA, and DEcircRNA target genes identified the function of protein processing in the endoplasmic reticulum, lysosome, and neutrophil-mediated immunity. Three subnetwork modules integrated from the PPI network also revealed the genes involved in protein processing in the endoplasmic reticulum, lysosome, and protein degradation processes, which may regulate the Xp11 translocation RCC process. The ceRNA complex network was created by Cytoscape, which included three upregulated circRNAs, five upregulated lncRNAs, 24 upregulated mRNAs, and two downregulated miRNAs (hsa-let-7d-5p and hsa-miR-433-3p). The genes as a prominent component of the complex ceRNA network may be key factors in the pathogenesis of Xp11 translocation RCC. Our findings clarified the genomic and transcriptional complexity of Xp11 translocation RCC while also pointing to possible new targets for Xp11 translocation RCC characterization.

## 1. Introduction

An uncommon and distinct type of kidney cancer, called Xp11 translocation renal cell carcinoma (RCC), is defined by TFE3-related gene fusions that occur on chromosome Xp11.2 [1,2,3,4]. The categorization of renal tumors by the World Health Organization (WHO) in 2004 was the first to formally acknowledge it [4,5]. Although just 1.6–4% of RCCs in adults have TFE3-rearranged mutations, this percentage rises to 40% in pediatric RCCs [6,7,8,9]. Xp11 translocation RCC has more aggressive clinical and pathological characteristics upon diagnosis and a poorer prognosis when compared to other RCC subtypes [3,5]. Unfortunately, the best treatment for Xp11 translocation RCC is still up in the air because of the limited knowledge of the probable processes and molecular characteristics [3], and a deeper comprehension of the molecular basis of the process is essentially needed.

Competing endogenous RNAs (ceRNAs), which methodically functionalize miRNA response element-harboring non-coding RNA and create intricate miRNA-mediated ceRNA networks, provide a significant layer of gene regulation by competing with one another for shared microRNAs [10,11,12,13]. CeRNA crosstalk disruption can upset the harmony of cellular developments and activities, resulting in the emergence of illnesses such as cancer [10,14]. Non-coding RNAs, including microRNA (miRNA), circular RNAs (circRNA), and long non-coding RNA (lncRNA), are also well established to have a significant influence in cancer courses of pathology and physiology [14,15,16]. Through functioning as tumor suppressors and oncogenes, miRNA expression dysregulation is correlated with a variety of malignancies [15]. LncRNA controls the transcription and post-transcriptional stages of gene expression, particularly by functioning as miRNA sponges in the ceRNA network [14]. In both physiological and pathological processes, the deregulation of circRNA is described in recent findings. Moreover, circRNA specifically controls gene expression by competing with ceRNA [16]. Data of all non-coding RNA and mRNA can be obtained by whole transcriptome sequencing (WTS) [17]. However, WTS strategies have rarely been used in delineating the transcriptomic panorama of Xp11 translocation RCC, which can reveal critical gene pathways to cancer.

Five RCC patients with Xp11 translocation participated in this work, and WTS was carried out on the tumor tissues (RCC) and their matching adjacent normal tissues (Normal). Subsequently, we applied differential analysis of messenger RNA (mRNA), miRNA, lncRNA, and circRNA between the RCC and Normal groups, and functional enrichment analysis is next. Finally, we constructed a complex ceRNA network by using miRNA regulation relationships and co-expression relationships of lncRNA-mRNA and circRNA-mRNA [10,11,12,13,14,17]. These hub genes may play fundamental roles in the pathogenesis of Xp11 translocation RCC and may be potential therapeutic targets in future studies. This study discovered molecular mechanisms potentially involved in tumor progression and may offer fresh perspectives on the molecular underpinnings of the Xp11 translocation RCC’s onset and progression.

## 2. Materials and Methods

### 2.1. Patients Identification

The research samples were taken from five Xp11 translocation RCC patients who received resection surgery at the Wuhan Union Hospital during the period 2016–2018. We treated these five patients with radical nephrectomy, and the postoperative pathological report showed the characteristic morphological appearance of Xp11 translocation RCC together with moderate to strong nuclear positivity, as seen by the findings of TFE3 immunohistochemical staining (TFE (+~+++)). Therefore, we diagnosed the patients with Xp11 translocation RCC. The study was managed following the Declaration of Helsinki and was approved by the Ethics Committee of Wuhan Union Hospital of Huazhong University of Science and Technology. Written permission for genetic analysis was obtained from all patients or their family members. Five couples from the RCC and the normal group performed the WTS to gain the transcriptome data. For sizable transcription sequencing and the categorization of non-coding RNA, whole transcriptome sequencing analysis is helpful.

### 2.2. Screening of the DEmRNA, DEmiRNA, DElncRNA, and DEcircRNA

Differential expression analysis of the mRNA, miRNA, lncRNA, and circRNA between the RCC and Normal groups was carried out by paired *t*-test method. |Fold Change| > 5 and level *p*-value < 0.001 were accepted as the cutoff values for evaluating significant DEmRNA, DEcircRNA, DElncRNA, and |Fold Change| > 2, and a *p*-value < 0.001 was believed to be the threshold for significant DEmiRNA. The heatmaps and volcano maps were drawn with the DEmRNA, DElncRNA, DEcircRNA and DEmiRNA by applying the pheatmap and ggplot2 R package.

### 2.3. Enrichment Pathway Analysis of DEmRNA

The widely used cluster Profiler R package [18] was used to analyze the Gene Ontology (GO) functional terms [19] and Kyoto Encyclopedia of Genes and Genomes (KEGG) pathways [20] in which DEmRNA was engaged. GO enrichment analysis is a technique for using the GO database to filter out significant, precise, and targeted gene functions that represent target gene groups. Finding the primary function of the trait carried by the target gene, determining the primary or secondary role of the same gene in the trait, and determining if the study target is correct or not over a bigger sample size are where its worth resides. KEGG is an effective instrument for metabolic network analysis and metabolic network research in vivo. The pathway enrichment analysis approach intends to annotate signaling pathways of the previously identified differential genes using the KEGG database and to retrieve all signal transduction pathways in which the genes participate. The significance level (*p* value) and misjudgment rate (FDR) of each signal route then were calculated using Fisher’s exact test and multiple comparison tests [21]. The enriched gene count ≥ 2 and the level *p* < 0.05 were judged meaningful for GO terms and KEGG pathways in this study.

### 2.4. Protein-Protein Interaction (PPI) Network and Module Construction of DEmRNA

STRING [22] was applied to investigate the interaction between DEG-encoded proteins, and we adjusted the parameter PPI score to 0.99 to find the most intimately connected interaction pairings. Cytoscape software (version 3.9.1) was used to create the PPI system net. Cytoscape’s plug-in MCODE [23] was utilized to investigate the PPI network’s most relevant clustering modules with the chosen score ≥ 8. GO enrichment analysis was also performed on the significant clustering module gene, of which the count ≥ 2 and level *p* < 0.05 were judged meaningful for GO terms.

### 2.5. Prediction of miRNA Regulation Relationship

The greatest database of predicted and empirically confirmed microRNA (miRNA)-target interactions is provided by the extensive repository miRWalk2.0 [24]. To investigate potential DEmiRNA-DEmRNA regulatory linkages, we retrieved 8 databases, including miRWalk, ENCORI, miRDB, RNA22, RNAInter, miRTarBase, TargetMiner, and TargetScan. We retrieved all the mRNA co-regulated by DEmiRNAs in these eight databases. To establish the regulatory links between DEmiRNA and DEmRNA, the predicted miRNA–mRNA interaction was merged with DEmRNA.

starBase2.0 contains the most extensive CLIP-Seq experimentally supported miRNA-mRNA and miRNA-lncRNA connection systems available for analysis [25]. Several confirming studies with stringency ≥ 1 were considered as the cutoff values for predicting the miRNA–lncRNA and miRNA–circRNA regulatory linkages of DEmiRNA by starBase2.0. The miRNA–lncRNA and miRNA–circRNA regulatory linkages were merged with DElncRNA and DEcircRNA to find the DEmiRNA–DElncRNA regulation relationship and DEmiRNA–DEcircRNA regulation relationship.

### 2.6. Co-Expression Analysis of DElncRNA-DEmRNA and DEcircRNA-DEmRNA

According to the matrix data of DElncRNA, DEcircRNA, and DEmRNA, we gained the Pearson correlation coefficients of each DEmRNA with DElncRNA and of each DEmRNA with DEcircRNA. Furthermore, meaningful mRNA–lncRNA and mRNA–circRNA couples were identified using a correlated test under the criteria |r| > 0.9 and a level *p* < 0.05. 

### 2.7. KEGG Enrichment Analysis of the DEmiRNA, DElncRNA, and DEcircRNA

According to the obtained DEmiRNA–DEmRNA regulatory linkages and the co-expression correlations discovered of DEmRNA–DElncRNA and DEmRNA–DEcircRNA, the cluster Profiler R package was applied for KEGG pathway enrichment analysis on the linked mRNA of DEmiRNA, DElncRNA, and DEcircRNA in these three couples. The analysis outcomes with count ≥2 and a level *p* < 0.05 were considered most significantly enriched. 

### 2.8. Construction of ceRNA Network 

The miRNA primarily attaches to mRNA at the post-transcriptional level to silence genes, while ceRNAs can function as a lure for miRNA attachment to counteract miRNA inhibitory action and establish ceRNA complex networks [26]. We combined the lncRNA–mRNA co-expression correlation and the DEmiRNA–DEmRNA and DEmiRNA–DElncRNA regulatory linkages. By the theory of ceRNA, we concentrated on the miRNAs that could control both DElncRNA and DEmRNA at the same time and promoted an mRNA–miRNA–lncRNA network.

In a similar way, we combined the circRNA–mRNA co-expression correlation and the DEmiRNA–DEmRNA and DEmiRNA–DEcircRNA regulatory linkages. By the theory of ceRNA, we concentrated on miRNAs that could influence both DEcircRNA and DEmRNA at the same time. There will be a deletion of DEmiRNA that is not exhibited in these two relationships to promote an mRNA–miRNA–circRNA network.

Moreover, we integrated these two complex networks. We concentrated on searching for miRNAs that can control DEmRNA, DElncRNA, and DEcircRNA simultaneously and the interactions that do not meet the criteria would be eliminated. We obtained a ceRNA complex network of co-expression correlation of DEmRNAs, DEmiRNAs, DElncRNAs, and DEcircRNAs. All these three relationship networks were created by Cytoscape [27].

### 2.9. GEO Data Validation

In this work, GSE167573 from the Gene Expression Omnibus (GEO) databases was applied to validate the results. Performing an unpaired *t*-test from the limma R package, the samples were put through a variance expression analysis to compare expression in tumors and normal tissues [28]. The criteria for screening substantially DEmRNA were |log2Fold Change| > 2 and a level *p* < 0.001. The Venn map was plotted by the DEmRNA obtained in this analysis and the results from GSE167573.

## 3. Results

### 3.1. Differential Expression Analysis

By the inclusion thresholds, 1886 DEmRNAs met the standards, including 1876, which were shown to be upregulated and 10 that were shown to be downregulated; 56 DEmiRNAs were found, and five were upregulated, and 51 were downregulated; 224 DElncRNAs were obtained in total, of which 216 were upregulated and eight were downregulated; a number of 1764 DEcircRNAs were identified, including 1760 that were upregulated and four that were downregulated. We plotted DEmRNA, DElncRNA, DEcircRNA, and DEmiRNA in four volcano maps and two-way clustering heatmaps (Figure 1A–D). As seen in these figures, we found that the main area of DEmRNAs, DElncRNAs, and DEcircRNAs were upregulated genes, and downregulated genes were dominant in DEmiRNAs.

### 3.2. Functional Enrichment Analysis of DEmRNAs

We conducted GO function and KEGG pathway enrichment analysis to further examine the biological function of the DEmRNA of XP11 translocation RCC. With GO function analysis, genes were enriched in 804 Gene Ontology Biological Process (GO-BP) terms, 209 Gene Ontology Cellular Component (GO-CC) terms, 119 and Gene Ontology Molecular Function (GO-MF) terms, and we summarized that DEmRNAs are mostly enriched in cell adhesion molecule binding, focal adhesion, cell-substrate adherens junctions, neutrophil-mediated immunity, and neutrophil degranulation (Figure 1E). As we conducted the KEGG pathway analysis, genes were enriched in 71 pathways, and we found that these DEmRNAs are mainly enriched in protein processing in the endoplasmic reticulum, amyotrophic lateral sclerosis, shigellosis, pathways of neurodegeneration-multiple diseases, and lysosome. Figure 1F displayed the top 12 pathways where DEmRNAs were enriched.

Going further, to explore the differences between the Xp11 translocation RCC and Kidney renal clear cell carcinoma (KIRC) and Kidney renal papillary cell carcinoma (KIRP), we processed differential expression analysis of messenger RNA (mRNA) in KIRC and KIRP separately from The Cancer Genome Atlas (TCGA) database. |log2Fold Change| > 2 and *p*-value < 0.001 were judged meaningful for the results (Figure 2A,B). GO function and KEGG pathway enrichment analysis revealed that the DEmRNAs of KIRC and KIRP are both mostly enriched in metal ion transmembrane transporter activity, monovalent inorganic cation transmembrane transporter activity, and neuroactive ligand–receptor interaction (Figure 2C–F). However, there were clear differences in enrichment results between KIRC, KIRP, and Xp11 translocation RCC. Transmembrane transporter-related function was insignificant in DEmRNAs of Xp11 translocation RCC. As shown in the Venn plot presented in Figure 2G, 18 DEmRNAs were found expressed in three cancers simultaneously, including SCARB1, APOC1, CALB1, SLC12A1, BHLHE41, ACSF2, GAS2L3, CD68, C3, HSPB8, UMOD, ALDOB, CGNL1, TIMP1, PKD1L2, NPIPB11, ANK2, and CYP3A5.

### 3.3. PPI Network and Module Construction

The DEmRNA-based PPI network included 1879 nodes and 1440 edges. Three subnetwork modules from the PPI network were integrated and retrieved with the Cytoscape plug-in MCODE (score ≥ 8) (Figure 3A,C,E). Module A (score = 29.655) with 30 nodes and 430 edges most prominently contained ribosomal proteins, such asRPL7A (degree = 29), RPS24 (degree = 29), and RPL32 (degree = 29). With 11 nodes and 55 edges, Module B (score = 11) was primarily made up of PSMB7 (degree = 10), PSMD8 (degree = 10), and PSMA6 (degree = 10). Module C (score = 8.133) had 16 nodes and 61 edges, the majority of which were NDUFS2 (degree = 10) and NDUFA6 (degree = 9). All these hub genes were upregulated.

Furthermore, a GO enrichment analysis was conducted on the hub genes in these three modules. The top 12 terms for every module were chosen for presentation in order of relevance (Figure 3B,D,F). The genes in module A highly contributed to protein targeting to ER, ribosome, and cytosolic part. Genes in module B were closely linked with the regulation of hematopoietic stem cell differentiation, regulation of the cellular amino acid metabolic process, and peptidase complex. Genes in module C were enriched in NADH dehydrogenase activity and the respiratory chain complex.

### 3.4. Enrichment Analysis of DEmiRNA, DElncRNA, and DEcircRNA-Related Target Genes

According to the mRNAs presented in the DEmiRNA-DEmRNA regulatory linkages and obtained DElncRNA-DEmRNA and DEcircRNA-DEmRNA co-expression correlations, Figure 4A–C displays the outcomes of the KEGG enrichment study, which demonstrated that target mRNAs related to DEmiRNA, DElncRNA, and DEcircRNA were highly enriched in protein processing in the endoplasmic reticulum, lysosome, and shigellosis. This result was consistent with the KEGG pathway enrichment analysis of DEmRNAs above.

### 3.5. Construction of ceRNA Network

Depending on the lncRNA-mRNA co-expression correlation and the DEmiRNA-DEmRNA and DEmiRNA-DElncRNA regulatory linkages, the miRNAs that do not regulate both lncRNA and mRNA will be eliminated. We identified 71 interaction pairs in total (Figure 5A), including six upregulated lncRNAs, 22 upregulated mRNAs, and three downregulated miRNAs.

Depending on the circRNA–mRNA co-expression correlation and the DEmiRNA–DEmRNA and DEmiRNA–DEcircRNA regulatory linkages, the miRNAs that do not influence both circRNA and mRNA will also be removed. Finally, 22 interaction pairs were obtained (Figure 5B), including three upregulated circRNAs, 17 upregulated mRNAs, and two downregulated miRNAs.

We combined the above two complex networks to filtrate DEmiRNAs which can regulate and control DEmRNA, DElncRNA, and DEcircRNA simultaneously. Eventually, 77 interaction pairings were discovered, including two downregulated miRNAs (hsa-let-7d-5p and hsa-miR-433-3p), three upregulated circRNAs, five upregulated lncRNAs, and 23 upregulated mRNAs (Figure 5C).

### 3.6. GEO Data Validation

Regarding the GSE167573 mRNA data, the filtering cutoff was chosen as *p* < 0.001, and |log2Fold Change| > 2, 4499 DEmRNAs were obtained in total. Additionally, we compared these DEmRNAs with the data gained in our analysis. As seen by the Venn plot diagram in Figure 6, 307/1886 (16.27%) DEmRNAs were shown to be differentially expressed in the GEO data. The variability presented in the GEO data findings could be related in some way to sample differences. Additionally, RPL12 with a greater degree in the enriched modules was among the 307 same DEmRNAs.

## 4. Discussion

As a rare and unique subtype of RCC, Xp11 translocation RCC demands special attention and more rigorous research due to its rarity, aggression in nature, and potential treatment possibilities [3,29,30]. WTS provides significant benefits compared to traditional sequencing technology, enabling most investigators to identify the tumor driver genes in a variety of cancers [17]. This study applied whole-transcriptome sequencing to identify 1886 DEmRNAs, 56 DEmiRNAs, 223 DElncRNAs, and 1764 DEcircRNAs in five groups of Xp11 translocation RCC samples. According to GO analysis results of the DEmRNAs, we discovered that DEmRNAs were mostly enriched in cell adhesion molecule binding, focal adhesion, cell-substrate adherens junctions, neutrophil-mediated immunity, neutrophil degranulation, and protein processing in the endoplasmic reticulum. PPI network analysis identified three sub-network modules, including ribosomal proteins, ribonucleoprotein, proteasome 20S Subunit family, and proteasome 26S Subunit family. KEGG pathway analysis on the linked mRNA of DEmiRNA, DElncRNA, and DEcircRNA also identified the function of protein processing in the endoplasmic reticulum, lysosome, and shigellosis. The downregulated DEmiRNAs, hsa-let-7d-5p and hsa-miR-433-3p, considerably stood out in the ceRNA network. Moreover, a few hub DEmRNAs were correctly verified by GSE167573; the discrepancy between the DEmRNAs and GSE167573 analysis results may be related to sample variances or a varying threshold selection.

Endoplasmic reticulum (ER) protein handling, modification, and folding are closely controlled activities that govern cell function, destiny, and survival. The ER homeostasis of malignant and stromal cells, also infiltrating leukocytes, is disrupted in a variety of tumor types by a combination of oncogenic, transcriptional, and metabolic aberrations that work together to create unfriendly microenvironments. These modifications cause ER stress, which has been shown to control a range of pro-tumorigenic characteristics in cancer cells while dynamically altering the activity of innate and adaptive immune cells [31,32,33]. Cancer cells stimulate the endoplasmic reticulum’s route for protein processing to achieve the proper protein synthesis and modifications and combat ER stress because of increased cellular metabolism and cell growth rates [34,35]. In a recent study, RNA sequencing was used to identify ribosomal protein-coding genes and proteasome genes engaged in the processes of RNA splicing and protein degradation in a variety of tumor tissues [13,36]. Additionally, there is growing evidence that suggests a connection between the neutrophil-to-lymphocyte ratio (NLR) and the survival of RCC patients receiving immune checkpoint inhibitors (ICIs) [37,38,39,40,41]. Several findings from this study stand out in especially, for example, the functional enrichment analysis of DEmRNA, and linked mRNA of DEmiRNA, DElncRNA, and DEcircRNA showed how protein processing in the endoplasmic reticulum and neutrophil-mediated immunity are connected. Moreover, the hub genes in three subnetwork modules contain ribosomal proteins, ribonucleoprotein, proteasome 20S subunit families, and proteasome 26S subunit families, such as RPL7A, RPL12, RPS24, RPS19, PSMB7, and PSMD8. These findings strongly suggest that genes which participate in the endoplasmic reticulum’s protein processing, neutrophil-mediated immunity, and protein degradation processes may be responsible for the etiopathogenesis of Xp11 translocation RCC, which offers a novel therapeutic concept for the identification of specific therapies for Xp11 translocation RCC in the future.

By controlling lysosomal biogenesis and autophagy, lysosome-mediated signaling pathways and transcription programs can monitor the state of cell metabolism and manage the transition between the metabolism of synthesis and decomposition [42]. Several studies have reported that some types of cancer, including renal cell carcinoma, are associated with overexpression of MiT-TFE genes, unifying how the lysosomal-autophagic pathway promotes the development and multiplication of cancer cells [43,44]. Moreover, lysosomes take part in additional procedures that encourage the growth of cancer cells. For example, cancer cells can have metastatic growth and invasion mediated lysosomal hydrolase exocytosis [45]. In our study, the lysosome appears in the results of KEGG pathway enrichment analysis on the DEmRNA and linked mRNA of DEmiRNA, DElncRNA, and DEcircRNA. Genes in module C were enriched in NADH dehydrogenase activity and respiratory chain complex. Furthermore, a recent study reported that lactate dehydrogenase B (LDHB) controls lysosome activity and autophagy in cancer. In the cancer cell, LDHB catalyzes the transformation of lactate and NAD (+) to pyruvate, NADH, and H (+) more than in normal tissue [46]. The lysosome may play an important role in the occurrence and development of Xp11 translocation RCC.

We also discovered that the hsa-let-7d-5p, which was downregulated, had a considerable impact on the ceRNA network. Numerous kinds of research have shown that hsa-let-7d-5p is a form of prognostic miRNA biomarker in various cancers, for instance, gastric cancer and esophageal cancer [47,48]. Moreover, overexpression of NEAT1 in cancer tissues has been proven to be associative with increased cell proliferation, migration, and invasion, as well as decreased cell apoptosis, according to a series of studies [49,50,51]. According to a research report, NEAT1 may be a significant mediator in the regulation of ccRCC progression and predicts the poor prognosis in patients with RCC [52]. The high expression of NEAT1 in RCC tissues predicts a worse five-year survival rate and cell proliferation was decreased and cell death was promoted in RCC cell cultures after NEAT1 knock-down by NEAT siRNA transfection. In research of targeted methylation of the lncRNA NEAT1, J. C. et al. reported that CRISPR/Cas13b-METTL3 methylation of NEAT1 promotes NEAT1 expression and indicates a tumor suppressor function in RCC [53]. It is also known that lncRNAs act as ceRNA by sponging miRNAs to alter the expression levels of their target genes in the development of cancers [54,55]. Plausibly, NEAT1 acted as a ceRNA of Xp11 translocation RCC in this study and may play a substantial part in the pathophysiology of Xp11 translocation RCC. The function of hsa-let-7d-5p and NEAT1 in Xp11 translocation RCC should be clarified by more future mechanistic research.

This study has several limitations. Firstly, because of the small incidence of Xp11 translocation RCC and its considerable heterogeneity, collecting an appropriate number of patients with distinct subtypes for further research is challenging. The limited sample number prevents our work from giving a full view of the XP11 translocation RCC transcriptome. Secondly, survival information was not available due to the short follow-up period. Thirdly, to define precise mechanics, more in vivo testing with animal models and functional descriptions is required.

In conclusion, our work performed whole-transcriptome sequencing to investigate the biochemical pathway of Xp11 translocation RCC and validated the results with GEO data. The genes engaged in endoplasmic reticulum protein processing and protein degradation pathways, such as RPL7A, RPL12, RPS24, RPS19, PSMB7, and PSMD8, as well as hsa-let-7d-5p, may be key figures in the pathophysiology of Xp11 translocation RCC. The integrated analysis of the ceRNA network also identified five hub lncRNAs, such as NEAT1, which has been proven to be associative with increased cell proliferation, migration, and invasion. Our findings contribute to the additional understanding of the genetic and transcriptional intricacy of XP11 translocation RCC and provide potential directions for further research. These genes could be considered in the future targeted therapies development for this cancer.

## Figures and Tables

**Figure 1 genes-14-00723-f001:**
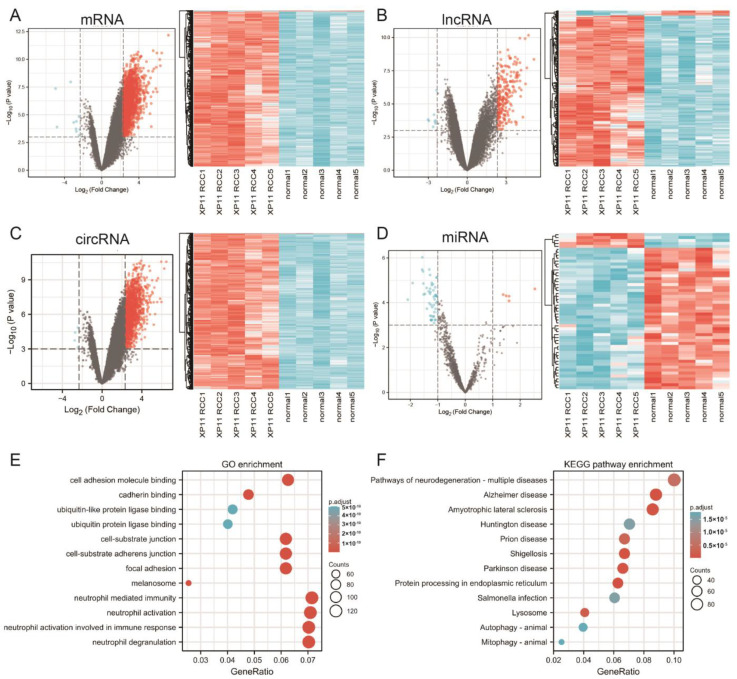
(**A**–**D**) Heatmaps and volcano maps of the DEmRNA (**A**), DElncRNA (**B**), DEcircRNA (**C**), and DEmiRNA (**D**). The red color represents upregulation, and the blue color represents downregulation. (**E**) Top 12 GO enrichment terms of DEmRNA. (**F**) Top 12 KEGG enrichment pathway of DEmRNA.

**Figure 2 genes-14-00723-f002:**
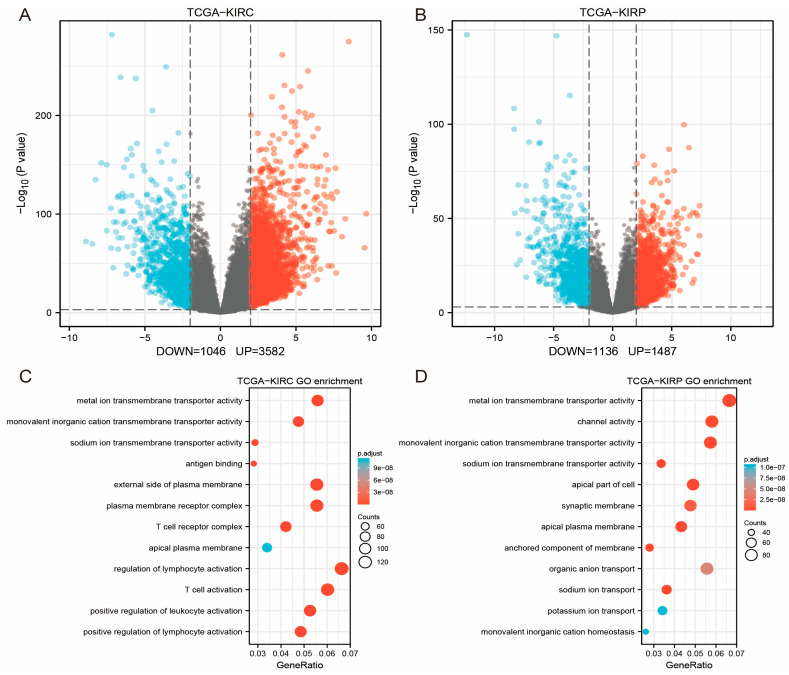
(**A**) Volcano maps of TCGA-KIRC. (**B**) Volcano maps of TCGA-KIRP. The red color represents upregulation, and the blue color represents downregulation. (**C**,**D**) Top 12 GO enrichment terms of TCGA-KIRC (**C**) and TCGA-KIRP (**D**). (**E**,**F**) Top 20 KEGG enrichment pathway of TCGA-KIRC (**E**) and TCGA-KIRP (**F**). (**G**) Venn plot of DEmRNAs of Xp11 translocation RCC, TCGA-KIRC, and TCGA-KIRP.

**Figure 3 genes-14-00723-f003:**
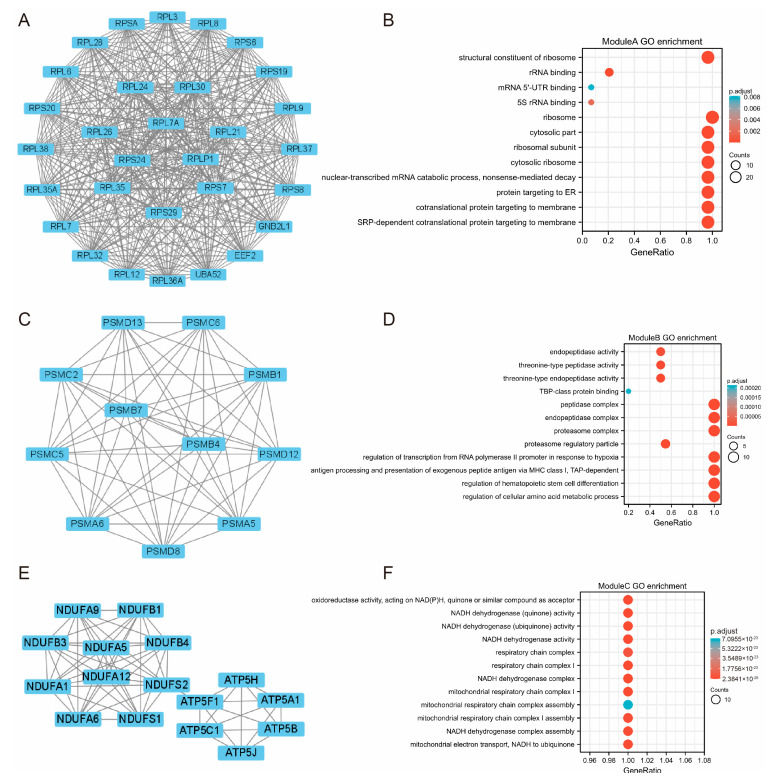
Three Modules Extracted from Protein-Protein Interaction (PPI) Network and GO Enrichment Analysis. (**A**) PPI network of module A. (**B**) Top 12 GO enrichment terms of module A. (**C**) PPI network of module B. (**D**) Top 12 GO enrichment terms of B. (**E**) PPI network of module C. (**F**) Top 12 GO enrichment terms of module C.

**Figure 4 genes-14-00723-f004:**
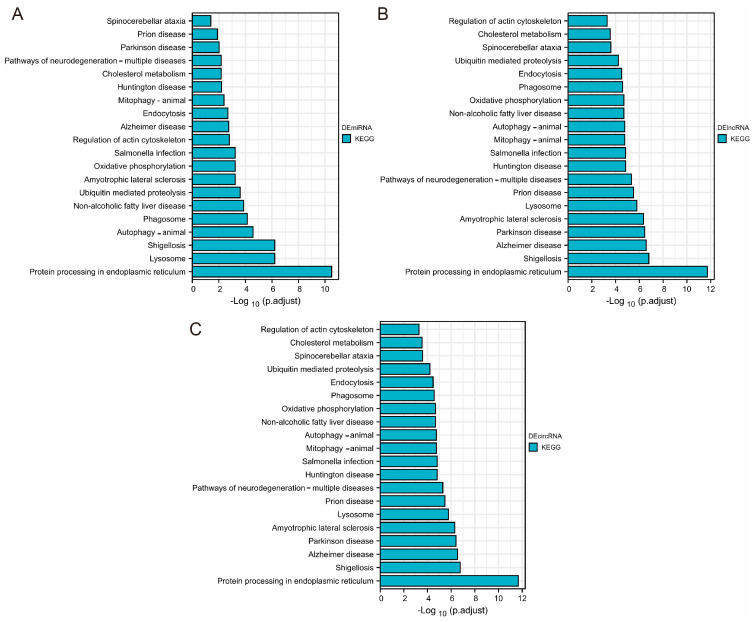
(**A**–**C**) Top 20 KEGG enrichment pathway of target mRNAs related to DEmiRNA (**A**), DElncRNA (**B**), and DEcircRNA (**C**).

**Figure 5 genes-14-00723-f005:**
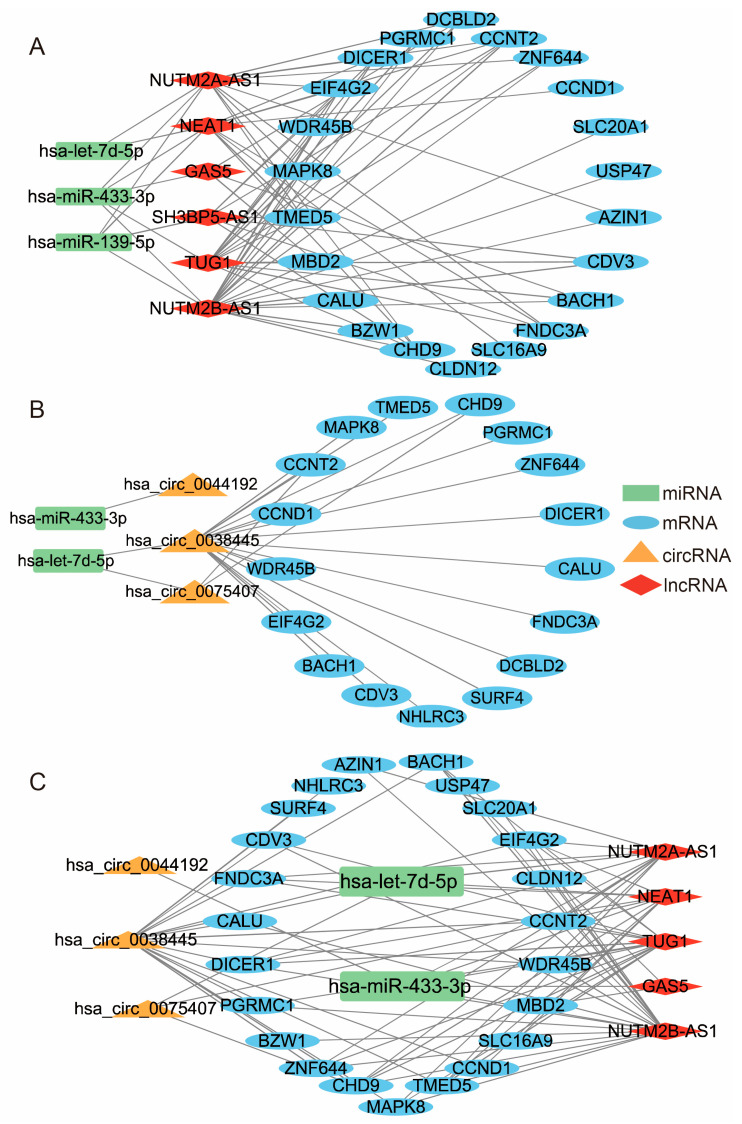
(**A**) The lncRNA–miRNA–mRNA network. (**B**) The circRNA–miRNA–mRNA network. (**C**) The ceRNA network.

**Figure 6 genes-14-00723-f006:**
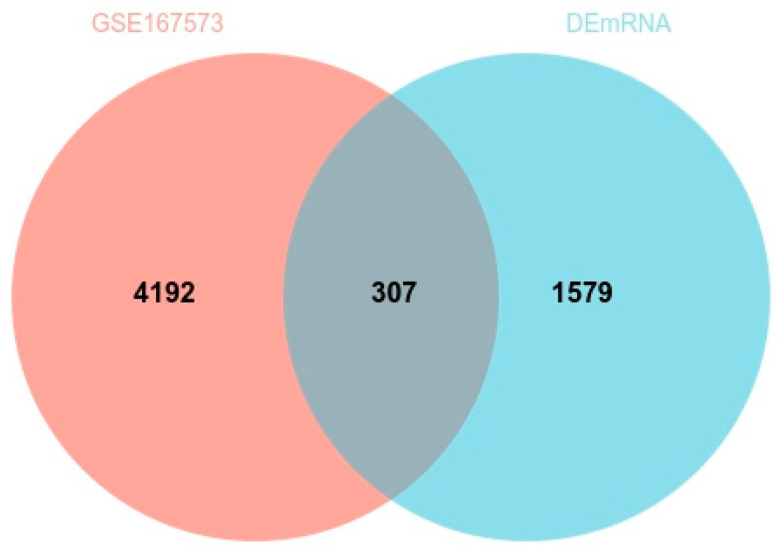
Venn plot of DEmRNAs of GSE167573 and this study.

## Data Availability

Data supporting reported results can be found in the Appendix A.

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
