# Peer review of "Identification of Key Differentially Expressed mRNAs, miRNAs, lncRNAs, and circRNAs for Xp11 Translocation Renal Cell Carcinoma (RCC) Based on Whole-Transcriptome Sequencing"

_genes, 2023, doi:10.3390/genes14030723_

Round 1

Reviewer 1 Report

The authors present a study describing whole transcriptome sequencing on the Xp11 translocation RCC. This study is comprehensive and adds to current knowledge on this matter. The major limitation is verbal small sample of patients. Additional questions could be addressed:

Q1: How was pathologic diagosis of Xp11 translocation RCC made?

Q2: How could your results potentially influence clinical practice in treatment of thesee patients?

Author Response

Response to Reviewer 1 Comments

Point 1 How was pathologic diagnosis of Xp11 translocation RCC made?

Response: Thank you for your comments. We treated these five patients with a radical nephrectomy and the postoperative pathological report showed the characteristic morphological appearance of Xp11 translocation RCC together with moderate to strong nuclear positivity as seen by the findings of TFE3 immunohistochemical staining. Therefore, we made the diagnosis of Xp11 translocation RCC in the postoperative pathology report.

Point 2 How could your results potentially influence clinical practice in treatment of these patients?

Response: Thank you for your comments. In this study, we found that the genes engaged in endoplasmic reticulum protein processing and protein degradation pathways such as RPL7A, RPL12, RPS24, RPS19, PSMB7 and PSMD8 may play fundamental roles in the pathogenesis of Xp11 translocation RCC. The integrated analysis of the ceRNA network also identified five hub lncRNAs such as NEAT1 which has been proven to be associative with increased cell proliferation, migration, and invasion. Our findings shed light on the genomic and transcriptional complexity of Xp11 translocation RCC while also pointing to possible new targets for Xp11 translocation RCC characterization. These genes could be considered in the future targeted therapies development for this cancer.

Reviewer 2 Report

The paper named “Identification of Key Differentially Expressed mRNAs, miRNAs, lncRNAs, and circRNAs for Xp11 translocation renal cell carcinoma (RCC) based on Whole-Transcriptome Sequencing” carried out a whole transcriptome sequencing (WTS) on the tumor and the matching adjacent normal tissues from five patients of Xp11 translocation renal cell carcinoma (RCC) in order to obtain more understanding of the genomic panorama and molecular basis of this cancer type. Authors found important information about genes involved in this cancer and give ensigns about molecular mechanisms potentially involved in tumor progression.

Only minor questions are required

1)      In material and methods point 2.2 authors give different cutoff values different for evaluating the genes significantly deregulated in DEmRNA, DEcircRNA, DElncRNA and in DEmiRNA. Why use different cutoff?

2)      Figures caption are missing

3)      In figure 1 it is impossible to see the heatmaps or the volcano plots. The figures are so smalls and the letter is impossible to read

4)      In line 171 the volcano plots maps do not mach with figure 1

5)      In figure 2 C D E and F is too small, the same as before, is difficult to see the data

6)      The Venn diagram that is shown in figure, I think figure 4 is perhaps more interesting if they are change to the final of the figure because this data are mentioned in the last point

7)       And part E, F, G is impossible to see

8)      This results, although they are very interesting are only related to genes and only functional analysis and comparison between the different samples but a PCR to validate some of the interesting result must increase the interest of the work.

Author Response

Response to Reviewer 2 Comments

Point 1 In material and methods point 2.2 authors give different cutoff values different for evaluating the genes significantly deregulated in DEmRNA, DEcircRNA, DElncRNA and in DEmiRNA. Why use different cutoff?

Response: Thank you for your comments. Due to the limitations of the website STRING, we must keep the number of proteins in the PPI network within 2,000. So we set |Fold Change| > 5 and level p-value < 0.001 as the cutoff values for evaluating significantly DEmRNA, DEcircRNA, DElncRNA. However, very few miRNAs were screened under this condition. In order to obtain sufficient miRNA for analysis, we set a different cutoff.

Point 2 Figures caption are missing.

Response: Thank you for your comments. Figure legends have been detailed in the manuscript.

Point 3 In figure 1 it is impossible to see the heatmaps or the volcano plots. The figures are so smalls, and the letter is impossible to read.

Response: Thank you for your comments. All Figures in the manuscripts have been modified accordingly to make them legible.

Point 4 In line 171 the volcano plots maps do not mach with figure 1.

Response: Thank you for your comments. We have corrected the mistake.

Point 5 In figure 2 C D E and F is too small, the same as before, is difficult to see the data.

Response: Thank you for your comments. These Figures in the manuscripts have been modified accordingly to make them legible.

Point 6 The Venn diagram that is shown in figure, I think figure 4 is perhaps more interesting if they are change to the final of the figure because this data is mentioned in the last point.

Response: Thank you for your comments. We have changed the Venn diagram in the Figure 4 to the final Figure 6.

Point 7 And part E, F, G is impossible to see.

Response: Thank you for your comments. These Figures have been redrawn to see clearly.

Point 8 This results, although they are very interesting are only related to genes and only functional analysis and comparison between the different samples but a PCR to validate some of the interesting result must increase the interest of the work.

Response: Thank you for your comments. I am sorry that the experiment involved was difficult to implement due to the limited number of samples the time of revision. Cell samples from the sequenced patients in this study have been difficult to collect again. Thanks again for your comments. We will follow your guidance and continue to explore in future studies to further verify the conclusion with new tumor samples and compare it with the current study.

Reviewer 3 Report

The article “Identification of Key Differentially Expressed mRNAs, miR-2 NAs, lncRNAs, and circRNAs for Xp11 translocation renal cell 3 carcinoma (RCC) based on Whole-Transcriptome Sequencing” by Deng et al. 2022 has merit.  I have a few concerns i.e.,

1.   Kindly conclude the abstract more deeply such as how this study will be useful for the society.

2. Can add the following reference: 10.1021/acsomega.2c01820 at methodology section “2.3. Enrichment Pathway Analysis of DEmRNA”.

3. Figure 2C 2D 2E 2F can be written as Figure 2C-F and be maintained for all Figures in the manuscripts.

4. Figure legends are missing in page no. 5

5. In Line no. 38-40, 55-57, 58-64 reference should be added.

6. Figure legends should be clearly mentioned

7. Images of the MS should be improved.

8. Limitation of the study should be clearly mentioned

9. Check Grammarly in the whole Ms.

Author Response

Response to Reviewer 3 Comments

Point 1 Kindly conclude the abstract more deeply such as how this study will be useful for the society.

Response: Thank you for your comments. We have modified the abstract according to your suggestion.

Point 2 Can add the following reference: 10.1021/acsomega.2c01820 at methodology section “2.3. Enrichment Pathway Analysis of DEmRNA”.

Response: Thank you for your comments. We have added this reference in the article.

Point 3 Figure 2C 2D 2E 2F can be written as Figure 2C-F and be maintained for all Figures in the manuscripts.

Response: Thank you for your comments. All Figures in the manuscripts have been modified accordingly.

Point 4 Figure legends are missing in page no. 5.

Response: Thank you for your comments. We've rearranged the Figures layout.

Point 5 In Line no. 38-40, 55-57, 58-64 reference should be added.

Response: Thank you for your comments. We have added reference notes where appropriate.

Point 6 Figure legends should be clearly mentioned.

Response: Thank you for your comments. Figure legends have been detailed in the manuscript.

Point 7 Images of the MS should be improved.

Response: Thank you for your comments. We have recreated the images with higher quality.

Point 8 Limitation of the study should be clearly mentioned.

Response: Thank you for your comments. Limitation of the study have been restated clearly in the discussion section of the manuscript.

Point 9 Check Grammarly in the whole Ms.

Response: Thank you for your comments. We have checked all grammatical errors in the manuscript and corrected them.

Round 2

Reviewer 1 Report

No further comments